# Community health intervention through musical engagement (CHIME) in South Africa: A formative exploration of the feasibility and development of a music-based intervention to support perinatal mental health

Siphumelele Sigwebela[1], Bonnie B. McConnell[2], Ncumisa Waluwalu[3], Thandi Davies[4], Katie Rose M. Sanfilippo[5]*, Lauren Stewart[6], Sally Field[3], Vivette Glover[7], Simone Honikman[3]

1 Institute For Life Course Health Research, Stellenbosch University, Stellenbosch, South Africa, 2 School of Music, Australian National University, Canberra, Australia, 3 Perinatal Mental Health Project, Alan J Flisher Centre for Public Mental Health, Department of Psychiatry and Mental Health, University of Cape Town, Cape Town, South Africa, 4 Alan J Flisher Centre for Public Mental Health, Department of Psychiatry and Mental Health, University of Cape Town, Cape Town, South Africa, 5 School of Health & Medical Sciences, City St George's, University of London, London, United Kingdom, 6 School of Psychology, University of Roehampton, London, United Kingdom, 7 Institute of Reproductive and Developmental Biology, Imperial College London, London, United Kingdom

* katie-rose.sanfilippo@citystgeorges.ac.uk

## Abstract

In South Africa, perinatal depression, stress or anxiety affect an estimated 16% to 50% of women posing serious concerns for both mothers and infants. The vast majority of women receive no perinatal mental healthcare through the public health system, partly due to high levels of stigma and a lack of culturally sensitive mental health care. South African musical traditions such as group singing are culturally significant for supporting social connection and coping with challenges experienced in everyday life. However, there is little research on how group music making could be used to support perinatal mental health in South Africa. This study aimed to explore the potential for developing a culturally embedded, music-based intervention to support women in the perinatal period. Using Community-Based Participatory Research, we held five focus group discussions with: 1) community health workers, 2) music experts, 3) traditional healers, 4) professional healthcare workers, and 5) the management team of a rural health NGO. Through thematic analysis, four themes were identified. Theme 1 encompasses the various challenges that contribute to perinatal mental distress, including social determinants of mental health, unhelpful coping strategies, stigma, and isolation. Theme 2 reflects existing community music practices: the way music is embedded in culture, processes of cultural change, and musical practices associated with perinatal health. Theme 3 encompasses the perceived benefits of music making in supporting social connections

**Data availability statement:** The focus group discussion transcripts hold personal information and stories and therefore cannot be shared publicly. De-identified data that correspond to the results reported in this article may be made available only upon reasonable request from the Perinatal Mental Health Project (PMHP), where the data is held. To request the data, you can contact PMHP Founder & Director Assoc. Prof Simone Honikman (simone.honikman@uct.ac.za). You can also contact the chair of the Human Research Ethics Committee, Faculty of Health Sciences at the University of Cape Town at hrec-enquiries@uct.ac.za.

**Funding:** This work was supported by a South East Network for Social Sciences/The Economic and Social Research Council-funded post-doctoral fellowship awarded to KRMS (Grant Reference Number: ES/V010158/1) and a grant from Goldsmiths University of London Strategic Research Fund awarded to LS. The funders had no role in study design, data collection and analysis, decision to publish, or preparation of the manuscript.

**Competing interests:** The authors have declared that no competing interests exist.

and effecting transformation in relation to individual mood and spiritual experiences. Theme 4 includes consideration of factors that are important for the development of a music-making intervention to support perinatal mental health. The findings suggest strong potential for implementing music-based mental health interventions in South Africa, adaptable to various facilitators and community contexts.

## Introduction

Stress, anxiety and depression during pregnancy and the first year after the birth of a child (perinatal period) affect not only the mother but can also have long-term adverse effects on her child [1–3]. In much of Africa, perinatal mental health services are minimal or non-existent, and high levels of stigma associated with mental health issues impede recognition and prevent help-seeking behaviour [4–9]. In South Africa, prevalence of depression during pregnancy is reported to be 22–47%, and 11–34% during the year postpartum [10–12]. It is thus of high priority to develop low-cost, low-resource, non-stigmatising and culturally appropriate approaches to reduce symptoms of anxiety and depression during this vulnerable stage in the life course. Research in high income countries has shown that singing in groups can be a powerful modulator of mood and emotion, reducing symptoms of anxiety and depression, increasing well-being, social affiliation and group bonding, including in the perinatal period [13–18]. However, previous research in community mental health and perinatal mental health interventions in Low- and Middle-Income Countries (LMIC) focuses on educational and psychosocial interventions primarily delivered via conversations and social gatherings [19,20]. The use of group music-making as the vehicle to deliver a perinatal mental health intervention has been studied in The Gambia by some members of our research team (BM, KRMS, LS, VG) and others, as part of the original CHIME project (Community Health Intervention through Musical Engagement for perinatal mental health) [21,22].

Through collaboration with women's community music groups (Kanyeleng groups) as the co-developers and facilitators of a perinatal mental health intervention, The Gambia CHIME intervention aimed to create a socially supportive network and lift pregnant women's mood, through group music-making [22]. The intervention sessions lasted one hour and were held weekly over six weeks. They were delivered to groups of about 10 pregnant women with no preselection. During each session, the Kanyeleng groups introduced specific songs drawing on traditional repertoire but adapting them to include new lyrics focused on agreed messages around (a) common physical and psychological symptoms of pregnancy, (b) techniques to cope with and manage these, (c) the importance of the CHIME group and other positive relationships in providing support, (d) the importance of being open and removing stigma to discuss challenges and promote empowerment, and (e) select messages on childcare. The participants were encouraged to join in by singing, moving to the music and clapping. Sessions involved call-and-response singing, with participants improvising and singing along. Each session began with a welcome song and ended

with a closing song. One lullaby was also introduced at each session to give the women repertoire to draw on after birth. A feasibility trial found that the CHIME intervention was feasible, enjoyable and culturally acceptable, and furthermore that it resulted in a significant reduction in symptoms of anxiety and depression in those receiving the intervention compared to those who received standard care [21].

South Africa has a rich history of community-based music practices and oral tradition. These are drawn upon especially during times of hardship, as a tool for socio-political and philosophical commentary, for the processing and reflection of historical moments and for building social cohesion [23–26]. The main ethnic groups (Nguni, Sotho, Tswana, Tsonga and Venda) each have their own varied and distinct cultural customs and traditional music, with a shared, common characteristic being the use of songs and instruments to accompany dance [24,25].

Traditional healers are considered critical health and spiritual guides across indigenous groups in South Africa. For many practitioners, the traditional healing process, whether in practice (for oneself) or consultation (for others), is facilitated through music making, vocal chanting, synchronised clapping and drumming [24,27–29]. In South African rural and semi-urban communities, traditional healers are considered primary consultants for mental health treatment, due to their accessibility, affordability and holistic and culturally sensitive approach [30–32]. South Africa's socially significant music culture and music making traditions suggest that here there is a strong potential to employ music to support perinatal mental health, informed by the CHIME approach.

The research team collaborated with One to One Africa, a Non-Governmental Organisation (NGO) in Mankosi, Eastern Cape, where local women deliver maternal and child health services as community health workers (CHW) or Mentor Mothers (MM). These women have developed an existing spontaneous musical engagement practice to deliver public health messages within their day-to-day work [33].

The goal of this project was to develop a Community Health Intervention through Music Engagement (CHIME) to support perinatal mental health in South Africa, building on the existing musical practice of CHW and the local community in the Eastern Cape, and informed by the approach co-developed in The Gambia [21]. The formative phase forms the basis of this paper. The co-development phase forms the basis of a subsequent paper in development.

In the formative investigation, we aimed to understand the priorities and needs of perinatal women in the community, gauge whether a CHIME-type intervention in the South African context would be appropriate, what content and process the intervention may have for optimal impact and accessibility and how the co-design process may best be conceived.

We consulted with key stakeholders to integrate existing local knowledge and practices related to maternal and mental healthcare with participatory music-making. This paper will describe the process and present the results of the formative investigations with stakeholders.

## Materials and methods

### Study design

Community Based Participatory Research (CBPR) is a research approach that emphasises shared power and participation between the community, user and researcher. CBPR is centred in nurturing connections within the targeted community in order to develop a network for the community to sustain interventions [34,35]. A critical methodology used in this approach is to engage with key informants throughout the development and implementation of interventions. The main outcome of CBPR aims to ensure the target community has support through the built network of stakeholders, and can be empowered to take action [36,37]. This approach was selected to ensure that the final intervention is culturally relevant and effective. The CBPR approach employed in this project built upon a longer-term partnership between the lead researcher (SH) and the partner organisation in the Eastern Cape, and was grounded in the existing strong musical practice of CHW within the community. The CBPR approach recognised the importance of cultural and musical expertise and traditional health knowledge systems as well as medical expertise. Focus group discussions (FGDs) were used to explore

and synthesise relevant insights from diverse expert stakeholders and community members during the first stage of the research.

CBPR has been criticised for the absence of a theoretical framework guiding the processes [38], prompting a move to synthesise CBPR into social movement theoretical frameworks and feminist theories that the principles of the CBPR methodology are drawn from. As such, this research study adopts a decolonial feminist theoretical perspective through the research study methods and analysis. To emphasise the aspect of empowerment in CBPR and maintain the decolonial approach to power, the context of the research is centred on the voice and perspectives of the 'marginalised' [39,40]. The decolonial feminist approach was evident in the way that women's community-based cultural knowledge and expertise was engaged as the foundation of the research and the starting point for intervention development. The methodology was designed to ensure that participants without prior experience of participating in research (including participants with limited formal education) could contribute actively and equitably. This paper reports only on the formative stage of the work, which informed the subsequent community-based co-design process with CHW. We plan for the co-design process as well as a description of implementation to be the subjects of a future paper. The Consolidated Criteria for reporting qualitative research (COREQ) was used and the COREQ checklist can be found in S1 Checklist.

## Study sites and sample

Five stakeholder groups with 28 participants were included for the FGDs (see Table 1). Snowball and convenience sampling [41,42] in the research team's personal networks were used to recruit music experts, traditional healers, and healthcare professionals for the first three FGDs. Participants were invited via email, telephone, and face-to-face. Our primary investigator (SH) built on a prior collaboration with the NGO, to engage with their management staff and CHWs for a further two FGDs. The NGO site, like others in the largely rural Eastern Cape province, is characterised by high levels of poverty, unemployment and limited infrastructure. Health services are under-resourced and access is constrained by distance and challenging terrain [43]. The province presents a culturally rich and community-rooted context in which community-based, participatory interventions may have high relevance.

## Procedures

The FGDs were conducted between 22 January 2022 and 28 February 2022 by a varying combination of two members of the research team. All facilitators had experience and training in FGD facilitation and experience in mental health research. The FGDs lasted between 90–120 minutes and followed a semi-structured guide (S1 Text) with opening

**Table 1. Categories of stakeholders interviewed in the formative investigation and the research facilitator from the CHIME team.**

| Group | n | Description | Location (platform and province) | Facilitator |
|---|---|---|---|---|
| NGO Management | 3 | The management team from the NGO, including a former CHW, a site manager and the head of programmes | Online (Eastern and Western Cape) | SH & SS |
| Music Experts | 5 | Music experts, including postgraduate music students, senior lecturers, professors and musicians | Online (Western Cape, KwaZulu Natal, Limpopo, Gauteng) | SH, TD& SS |
| Community Health Workers | 7 | Nominated by the NGO out of a cohort of over 30 CHW; these women are also musicians, composers and poets. | In person (Eastern Cape) | SS & NW |
| Traditional Healers | 10 | From the Union of Traditional Healers in the Western Cape. These healers are herbalists, prophets and a medical and traditional midwife | In person (Western Cape) | TD & SS |
| Healthcare Professionals | 3 | Clinical psychologist, provincial lead at Western Cape Department of Health (Professional Nurse), Public Health, Behavioural Change & Community Change Specialist (Medical doctor) | Online (Western Cape, Gauteng) | SH & TD |

questions and probes. These guides were adapted and tailored to the expertise of each stakeholder group. However, each semi-structured focus group aimed to answer the following research questions:

1. What community music practices already exist?

2. How might they intersect with perinatal mental health care?

3. Do stakeholders think a community-based music intervention would be helpful for women during pregnancy and after birth, and how?

4. What key perinatal mental health messages could be communicated in this form?

5. In what form might participatory music making be used in a perinatal mental health intervention?

6. What co-design approach may be used to prototype a repertoire of musical engagement strategies to promote perinatal mental health?

For the traditional healers and CHWs, FGDs were conducted in isiXhosa, their first language, by isiXhosa speaking researchers. All other FGDs were conducted in English by English-speaking researchers as this was well understood or the first language of the other participants. All FGDs were audio or video recorded, and field notes were made by facilitators. After each FGD, the whole research team met to discuss the data.

## Ethical considerations

All participants were over 18 years old and provided written informed consent. The protocol received ethics approval from The Health Sciences Faculty Research Ethics Committee at the University of Cape Town (ref: 750/2021) and Goldsmiths, University of London Research Ethics & Integrity Committee (ref: 1594). Online participants who did not have access to free internet connection were provided with data vouchers. In-person participants were provided with transport to the venue, refreshments and a grocery voucher valued at ZAR (R200) ($10) for each day of their participation, to compensate for their time.

## Positionality of the research team

This study is unique in bringing together a highly interdisciplinary team with expertise across the area of psychology, psychiatry, public health, and ethnomusicology. Our collaborative, interdisciplinary approach sought to overcome the problems associated with the disciplinary siloing of research on music and mental health [44]. The research team was composed of researchers from South Africa, United Kingdom, Australia and the United States. We took steps to mitigate the risk of perpetuating historical power imbalances in our research. Our goal was to foster a more equitable and inclusive research environment that respects the diverse perspectives of all participants [35]. Careful consideration was taken into selecting FGD facilitators according to the context and the needs of each FGD stakeholder group (see Table 1). Facilitators were selected based on their research expertise and their proficiency in language to build trustworthiness between participants and the research team. The facilitators clearly introduced the project, their own background and motivation for the research prior to each focus group discussion. Co-facilitators were matched according to experience to minimise hierarchies during facilitation and to develop the capacity of the junior research team members. This enabled the facilitators to deviate from the formal style of academia and reflect on their own commonalities and biases during the discussion, enabling sessions to become fluid and conversational [45]. Along with extensive preparation with input from the senior members of the research team, and daily debriefing after each FGD, we utilised the knowledge of the research team and research participants to support the FGD facilitators. Prior to the start of data collection, the research team acknowledged their own biases within the context of the research, highlighting specifically their position in relation to the rural South African context (see S2 Text and S1 Table). Notably, SS, a young isi-Zulu speaking woman researcher, fluent in isi-Xhosa,

collaborated with a young isi-Xhosa speaking research assistant, NW, to conduct the focus group with the rural CHWs. Their young age relative to that of many of the CHWs informed their highly respectful and humble engagement approach with the participants. NW is also a mother, living within an adjacent community, which enabled her to draw on a nuanced appreciation of local practices and understandings - critical for the setup and conducting of the FGDs and for the analysis of the data.

## Data analysis

The FGDs were transcribed and translated by a trained isi-Xhosa PhD researcher in mental health, Independent of the research team and fluent in English. Two researchers (SS, BM) checked the transcription for any discrepancies. A thematic analysis [46,47] was conducted to identify categories and themes that could be used to represent the data. Thematic analysis allows for both inductive and deductive approaches of data interpretation and includes the following steps: familiarisation with the data, generating initial codes, searching for themes, reviewing these themes and defining and naming themes [48]. For the English-language FGDs, two researchers (SS, BM) independently identified categories to represent key elements of the data. Both researchers then came together to compare their codes, resolve any discrepancies and develop larger categories to be incorporated into a codebook. For the isiXhosa FGDs, two isi-Xhosa speaking researchers (SS, NW) generated codes for the original isiXhosa transcript; the two coders resolved any translation discrepancies by inserting the original isiXhosa text into the translated English transcript. This ensured that the integrity of what participants were saying was represented in the final transcript that was coded in English. The initial themes were shared with the NGO Management and the Traditional Healers for input. The themes were then shared with CHW participants for their feedback during the subsequent 3-day co-design workshop. This involved further discussion of the challenges relating to the design and delivery of a musical intervention for mental health in their community. The participants then synthesised the key themes for inclusion in a repertoire of songs and developed recommendations for a community-based intervention to support the perinatal mental health of mothers in their community (discussed in a separate publication under development). The analysis was completed using Dedoose [48], a digital tool for qualitative analysis. Once the codes were finalised, they were synthesised into broader themes to represent the data. The themes from the data analysis were used to describe the broader context of existing practices, and the implications of potential practices for mental health intervention development and traditional music-making in South Africa.

## Reflexivity

Reflexivity is critical to decolonial research and acknowledging the historical socio-political context of our research context was essential to both data collection and interpretation. Ongoing examination of positionality and bias was central to this work [35], supported by the regular group debriefings we held after each FGD and workshop to reflect on how our personal experiences shaped both data collection and interpretation. These reflections deepened our understanding of participants' experiences of trauma, cultural disconnection, and marginalisation [49,50]. This became especially important during the analysis of the traditional healers' FGD, where SS and NW's religious upbringing required careful self-awareness. This process directly informed the insights in Theme 2, particularly our acknowledgment of cultural disconnection and change, and the evolving purpose and form of healing music.

## Results and discussion

This research study aimed to explore the feasibility of a CHIME intervention in South Africa. Through FGDs with five stakeholder groups, several key themes emerged, including challenges related to social determinants of perinatal mental health, existing community musical practices, and music making to support mental health. The following sections will delve deeper into these themes and provide contextualised insights into the participants' experiences.

Participants engaged in detail on the psychosocial challenges that contribute to perinatal mental health problems. They had extensive experiences of community-based music practices, and many were familiar with how some of these practices supported mental wellbeing. The benefits of music making were well articulated and guidance was provided on how music making could be harnessed to optimise perinatal mental health. These findings were categorised in four themes, each including sub-themes, demonstrated in Fig 1. For more detail on the themes and sub-themes and example quotes see S2 Table.

**Theme 1: Challenges that contribute to perinatal mental distress**

This theme captures our participants' discussion of the perceived causes and contributing factors associated with perinatal mental health problems.

**Social determinants of poor mental health.** FGD participants described the hardships and problems that perinatal women experience. They explained that many women have problems with poverty, food insecurity, unemployment, HIV, gender based and sexual violence, other forms of abuse, abandonment and a lack of social support, which worsen during the perinatal period. One participant explained that these problems can be particularly acute for teenage mothers:

*"A teenager for real is the one that gives a hard time because as a matter of fact, others get pregnant because of these friends and doesn't even know what she will feed the child, but [she] got pregnant. There are not even the clothes… It is another thing now that makes her not to be alright… because she is thinking, 'What am I going to do with this child.'"* – CHW, woman

More than half the South African population lives in poverty, where access to water, food, electricity and housing is limited [51]. As one moves further away from the urban and peri-urban settings, these resources, including access to healthcare services or any formal mental healthcare, diminish further. The primary healthcare facilities that are available often provide for all the villages within a walking distance of 9km [43,52]. The group of women represented in our intended

**Theme 1:** **Challenges that contribute to perinatal mental distress**
- Sub-theme 1.1: Social determinants of poor mental health
- Sub-theme 1.2: Unhelpful coping strategies
- Sub-theme 1.3: Stigma and isolation

**Theme 2:** **Community music practices in South Africa**
- Sub-theme 2.1: Music is embedded in South African culture
- Sub-theme 2.2: Cultural change
- Sub-theme 2.3: Musical practices associated with perinatal health

**Theme 3:** **Benefits of music making**
- Sub-theme 3.1: Building social connection.
- Sub-theme 3.2: Transformation

**Theme 4:** **Making music to support mental health**
- Sub-theme 4.1: Ingredients of music
- Sub-theme 4.2: Contexts of music making

**Fig 1. Presenting Themes.**

intervention group come from remote and rural communities and so any existing problems are exacerbated by lack of access to care. These challenges are influenced by the racial and systemic oppression of people of colour experienced during Apartheid – the impact of which is sustained to the present day. Furthermore, inadequate resources have been allocated to the redevelopment of these communities. The lasting impacts of Apartheid, alongside African patriarchal norms [50] has led to persisting intergenerational trauma, dysfunctional family relationships, high levels of intimate partner violence and substance abuse rates in these communities [12,49,53,54].

**Unhelpful coping strategies.** Participants reported that pregnant women and mothers in their communities live with constant stress, with many also facing depression. Although there are some women who seek support from their families, friends, and church communities, participants explained that many women also manage their stress through unhelpful coping strategies such as avoidance, neglect, substance abuse and engaging in conflict.

*"I know it's happening generally everywhere, you find out that some, they get pregnant while they were not ready to be pregnant and now the partner is not there to give the kind of support. The woman herself is not in a state where she can be able to support herself. The woman will spend time thinking about, how am I gonna afford this child. So, now that…that leads to lots of stress now which in most cases you find out they end up… drinking while pregnant or either using drugs while pregnant because they are trying to avoid the stress and not having the alternative way"*– NGO management, woman

Moreover, participants described that some of these women have little-to-no support from family because their families do not understand that their behaviour is related to their distress. Therefore, it is important to acknowledge that unhelpful coping strategies can be a symptom of the environment that women and their families live in and that these systemic problems that exacerbate mental distress cannot be resolved by the efforts of mothers and their families alone. Indeed, one national study found alcohol use in pregnant women in South Africa was associated with being mixed race, unemployed, poor health status, and having Post-traumatic stress disorder , demonstrating links between systemic socio-economic issues and alcohol use [55].

**Stigma and isolation.** Participants described how perinatal women experiencing mental distress are often misunderstood, stigmatised and isolated, because there is a lack of education about mental health problems and mental distress. Below, a participant reflects on a common misconception about mental health that results in stigma:

*"A person will be walking and talking to herself and we will say she is a witch, only to find out she is affected mentally because she never got care by the time she was pregnant, and gave birth."* – CHW, woman

Miseducation about symptoms of depression and other common mental health conditions creates an environment for speculation and gossip, and the spread of misinformation. Often, the manifestation of mental ill health, through a woman's behaviour (even suicide in some cases) is blamed on others in the community, typically elderly women who are assumed to have caused the behaviour, through witchcraft. These accusations often result in the violence, abuse and death of these elderly women, which is an ongoing, endemic problem in rural South Africa [56,57]. Below, a participant describes the damage a lack of awareness about mental health can cause to women, highlighting the need for mental health education in their communities.

*"Here many people have died because of depression. We are not aware of the thing of depression. The thing that happens, the old women get killed because it is said that when I have completely lost my mind or hung myself, an elderly woman is accused… It's said she's the one who killed… Whereas it is depression; we don't take it seriously because we were never taught about it in the first place."* – CHW, woman

Overall, our analysis highlights the various challenges that women experience, including poverty, social isolation, stigma, and unhelpful coping strategies, that contribute to high levels of mental distress during the perinatal period.

These findings are consistent with mental health literature demonstrating the important role of social determinants in shaping health outcomes [58–60], while drawing attention to gendered aspects of marginalisation that have been under-explored in the literature. This provides important context for understanding priorities and needs, and the way a music-based intervention might be developed that is sensitive to both the individual challenges experienced by women and the broader social and cultural contexts shaping individual health outcomes. This approach is unusual in arts and health research, which has been dominated by research conducted in Western contexts and not always adequately considered the social and cultural context of interventions [61].

### Theme 2: Community music practices in South Africa

This theme reflects the rich historical and contemporary musical environment in many South African communities, perspectives on cultural change, and existing musical practices associated with perinatal health. Building on the insights developed in our discussion of Theme 1, in this section we examine the contexts of music making to provide a foundation for considering the potential role of music making as an intervention to support perinatal mental health.

**Music is embedded in South African culture.** Participants described how music and musicality play an important role in cultural life in South Africa. They explained how often music occurs when people gather together, and they highlighted the connection music has to spirituality within Black South African culture:

*"We have a natural relationship with music. If you look at our funerals, our weddings, our coming-of-age ceremonies and our traditional practices [there is always music]…." – Musician, woman*

The discussions also highlighted a resurgence of interest in ancestral practices linking music and spirituality, and for some participants this was described as a form of healing that is urgently needed in contemporary South Africa. For many of the participants, music was described as accessible, relatable, and able to break down barriers in interpersonal relationships.

In South Africa, there is an existing music culture based on social, cultural or religious beliefs. One participant described music as 'a phenomenon' because the technicalities of spontaneous music making are challenging to describe or give instruction for.

*"Africans don't talk about music, they just make music. As we are speaking now, they are singing there… Don't prepare what to teach them because they know what they are doing." – Music expert, man*

This quote relates to a broader tension that emerged in the discussions between formalised, Eurocentric notions of musical expertise (i.e., music theory and text-based knowledge) and embodied musical knowledge systems in Black South African culture. This points to the importance of considering power dynamics, notions of expertise, and locally produced meanings of music making in South African community contexts.

**Cultural change.** Participants explained that contemporary music is easy to access, but that some important forms of traditional music containing history and knowledge have become lost. This is a topic that has been addressed in research on South African music more broadly [26,62–64]. Participants also noted that in the past, women spent time in the fields, working the land, growing grains and sharing music together. One traditional healer described how modern society and its technological distractions, is not an ideal environment for healing music that previously contained messages of encouragement or useful knowledge about the roles and responsibilities of women, and how to navigate the world:

*"We are in the industrial days where women go to school, they don't go to the fields, so they don't know how to sing those songs… So, there is no time for those songs now, there is only time for TVs and loud music - that is not therapeutic for them, that has got an after effect for them."* – Traditional healer (herbalist), man

Our participants expressed concerns about the way access to forms of healing music for mental and perinatal health is dying with the elders and elder healers. Knowledge of music, including practices associated with health and healing, have traditionally been passed down orally by community or family elders, however, today much has changed [65]. As South African cultural groups strive to align with global society, many are adapting African traditions to a more Westernized global approach. In the process, traditional music has been modified and secularised, leading to a loss of some of its original meaning [65,66].There is a continued respect and regard for elders as family or community members, but their agency to contribute to day-to-day life has diminished [67]. They are seen as a group of people that need to be cared for or looked after, but not valued as people who have wisdom and can make meaningful contributions to adapting old ways with the new ones [68,69]. One participant described how knowledge of culture and music can become lost and the importance of engaging with elders and documenting their knowledge:

*"Now, when ugogo (grandmothers) have all these lullabies in her mind she passes away, all that is gone. You understand? So we need to keep records!"* – Traditional healer (prophet), man

The broad discussion of the way music making is embedded in South African culture, and concerns about loss of cultural knowledge, suggest that there is a need for new strategies to sustain musical practices into the future [70–72]. These perspectives are important in informing the approach to developing a music-based intervention for perinatal mental health. Particularly, they highlight the need to consider active participation and accessibility of music, as well as alignment with pre-existing cultural practices and norms.

**Musical practices associated with perinatal health.** Participants described varied examples of specific musical practices associated with perinatal health. These music practices, now less common than previously, occurred in hospitals, clinics and within families and communities. Participants described musical practices that occurred formally in hospital waiting rooms (as part of a student programme in a hospital). Taking an open approach to musical engagement, these practices were more generally aimed at upliftment and promoting cohesion between health worker and client. Participants also described musical practices for spiritual guidance using drums and singing, and music used for relationship building (for both staff and clients) and motivation. There are also existing long-standing musical practices related to cultural ceremonies or celebrations; each of the musical examples described below occurs within certain contexts, with specific practices used for a prescribed intention or to share a specific message.

Participants described participatory music practices common in the past that aimed to improve perinatal health and help women prepare for their antenatal clinic (ANC) appointments:

*"Back in the days, er…in the clinics each and every time when a pregnant woman comes in for their ANC appointments, they used to gather like in this one place and there, they would sing songs…that would prepare them to get into their consultation rooms like they would sing songs, songs that would also help with their stomach."*– NGO management team, woman

Participants also described how traditional midwives have contributed to perinatal healthcare for many generations. Their roles include being present at all stages of pregnancy and birth [73] where they teach songs to assist with labour. These songs are intended to help with breathing and managing the tension associated with labour and to help birth a baby safely.

*"We keep the people like senior female elder - the midwives who knows those songs - those are pre-labour songs we talked about, that you have to sing those songs; perform this process of giving labour to this child."* – Traditional healer (prophet), man

While in countries like Zimbabwe, Ghana and Ethiopia the role of the traditional midwife has been integrated into perinatal health care, in South Africa, there is little collaboration between traditional medicine practices and Western health systems [74]. Considering the knowledge that traditional midwives have on improving conditions for labour, including traditional music practices, their inclusion into primary health care would likely be beneficial for infant and perinatal outcomes in South Africa as well.

The current and historical precedents for inclusion of music during pregnancy, labour, and the postpartum period, as described in this section, suggest that there is potential to develop a culturally appropriate intervention to support perinatal mental health using group music making. While music research has long considered the important social and cultural roles of music making in South Africa [23–26], the potential of music to be engaged to support improved health outcomes during the perinatal period has been neglected in existing research. Following Sunderland et al.'s work with First Nations communities in Australia, we contend that community-based musical practices can serve as a "cultural determinant of health" [75] with particular significance for perinatal mental health in South African contexts due to the longstanding association of music, and singing in particular, with pregnancy and motherhood. In the next section, we examine the potential benefits of music making in more depth.

## Theme 3: Benefits of music making

Participants shared multiple psychosocial benefits of music-making, both in building social connection and transformation (of mood, stress, and spirit). These themes were also raised in relation to potential interventions to support perinatal mental health (Theme 4).

**Building social connection.** This sub-theme relates to how music may be used as a tool to rebuild and develop social connection.

Participants described how social connections are typically built through group activities such as gardening, exercising, crafting and music making. They mentioned that these group activities functioned to enable bonding and guidance through affirmation.

*"I think if we come up with activities that will bind them… Activities… anything that can guide them like singing"* – NGO Management, woman

Although many group music making activities are facilitated through cultural ceremonies, public and religious events, and family traditions, the need for an intentional group space for perinatal women was emphasised by many participants. In the excerpt below, a CHW imagined a support group that would be open to perinatal women in different phases, highlighting that the space should provide enjoyment or 'fun'. The original expression of fun in isiXhosa is synonymous with happiness, lightness, and joy, further emphasising the need for the group to feel separate from the hardships of ordinary realities.

*"As for me, I think one of the things that could help is for them to form a support group, those who are pregnant and the ones who have delivered. Where there will be fun [Apho kengoku kuzakonwatywa khona]"* – CHW, woman

Participants explained that fundamentally, music brings people together, and that this is more important than any specific messaging or intention of the music. They highlighted that the phenomenon of music making is at its core a shared experience, and that music is able to transform spaces and encourage emotional openness. Music making was also

described as an activity that connects and encourages people to share their experiences. One participant described how shared music can change a space and support people by helping them recognise that they are not alone and there is no need for shame. In her explanation, the isiXhosa word "*niphefumelana*" was used to describe the sharing of secrets and being open with one another to overcome shame:

*"Where you discover, my problem is smaller compared to… I thought mine was bigger. It is smaller because you breathe upon each other [niphefumelana], the other finds out that… it is bigger than mine. In so doing, each and every person feels free now and see that, 'No maarn, let me join the other kids, I'm not an embarrassment [andilohlazo maarn] I am the same as the other kids'".* – CHW, woman

The examples shared in the FGDs align with a larger body of research on music and music making as a means to facilitate social bonding and social connectedness [15,76]. This was identified as particularly important in the South African context for women experiencing stigma and marginalisation during the perinatal period.

**Transformation.** This sub-theme describes how music can serve as a tool to effect transformation of an individual's mood and spirit. Again, this sub-theme was discussed in relation to the affordances of music in general and more specifically in relation to the development of an intervention to support perinatal mental health.

In relation to individual mood, participants discussed the way that music making can transform negative emotional states. They noted that singing with others can serve to shift attention and allow people to focus on their experience in the moment, rather than ruminating on negative thoughts and feelings.

*"Meeting and singing those songs and thing also makes you to forget the thing you were thinking of while you were in the house, busy thinking… That meeting helps it because that thing tends to calm down that thing you had."* – CHW, woman

*"If there is singing and maybe listened to those words and how the singing is approached, it gets to my soul and no matter how much I was sad, I forget about anything."* – Traditional healer, woman.

Closely related to the previous point, participants also emphasised the way music making can transform anxious or agitated states by helping people to calm down, a theme that has also been addressed in the wider literature on music, health, and healing [77,78]. They noted that this may be experienced through listening to music, through singing to oneself, or (especially) through making music together with other people. One participant in the traditional healer focus group explained the importance of lullabies in calming not only the baby, but also the mother:

*"You are sitting there as the mother of the small child in a small room which is stuffy, you know like obviously you gonna get depressed. You see that thing? You know, so that is why our grandmothers… they invented the lullabies. So they said rather than get stressed here, it is better for me and my child to sing and sleep, you see! That is why they call it lullaby because it calms down your nerves, your stress, your depression; so that thing has been lost."* – Traditional healer, man

While benefits of lullabies were noted in our research, many participants expressed concerns about a loss of knowledge of lullabies as well as other song repertoires that are now known only by older people within the community. As discussed in relation to Theme 2, this suggests a need for new approaches to sustain song repertoires and music associated with the perinatal period.

In addition to the transformative effect of music on individual mood, the theme of transformation was discussed in relation to spiritual experience, trance, and traditional healing. This is also a theme that is emphasised in the broader literature

on music, spirituality and healing in South Africa [29]. Participants in the traditional healer's focus group noted the way that music can serve as the catalyst to awaken a person's spirit:

*"And they [the ancestors] tell you, "We want a song now" and you have to take that drum and beat it, or if there is someone who can beat that drum for you. A song is sung and the dancing happens at 1 o'clock, mid-night, and they would be surprised and feel like this person is mad, you see! That time your ancestors they want a song. You see, we are different. There are designated singers. There is someone who has that special … That when she sings your spirit is awakened."* – Traditional healer, man

A participant in the music experts FGD likewise emphasised the "power of music" in relation to traditional healing practices, suggesting that music itself "does not heal" but rather serves to transform a person's state in preparation for healing. These responses highlight an integral role of music in relation to spiritual experience and traditional healing practices [32,79].

Music's integral role in traditional healing practices in South Africa as well as in supporting social connection and support was strongly evident in our research and also acknowledged in the wider music research literature [27–30]. However, existing research has not considered the implications of these culturally significant musical practices for perinatal mental health. Nor has there been adequate attention to the challenge of musical sustainability in the context of cultural change. As referenced above, the work of Sunderland et al. [75] proposes that music making may play an important role as a cultural determinant of health for First Nations communities dealing with intergenerational trauma as a result of colonialism and racial oppression. In the South African context, we suggest that a culturally grounded music-based approach may be particularly important in addressing challenges of stigma and isolation (potentially both a symptom and cause of mental distress) discussed in relation to Theme 1. However, in the context of cultural change, our research suggests that there is a need for new sustainable, accessible approaches that engage music making as a cultural determinant of perinatal mental health. In the next section we examine in more detail the important musical and contextual features of such a music engagement intervention.

## Theme 4: Making music to support mental health

The sub-themes explored in this category involve participants describing the practicalities of a music engagement intervention. Participants emphasised the need to create a fun, comfortable space for music making that encourages active participation. Overall, the discussions suggest that creating a comfortable space for music making that is supportive of mental health depends upon the "ingredients of music" and the way they are experienced in context. Details included the procedure of potential sessions, including describing the location of such interventions as well as the characteristics of the music itself and ideal facilitators.

## Ingredients of music

This theme describes aspects of facilitating music making that should be considered to support mental health and well-being. Participants emphasised the need to identify forms of music that are accessible and familiar to people in the local context, rather than imposing unfamiliar musical styles or ways of musical engagement:

*"The most important thing would be to ask the group what they want to sing and what they want to do because we can't know what's going to be appropriate to your group, to your women and to their need unless we first find out what will they need, what music do they like, what would they like to sing? You know, we make many assumptions about, well if people are in the Eastern Cape that they are going to be like this."* – Musician, woman

More broadly, our discussions highlighted the need to consider music making in context, rather than assuming all music would be beneficial for mental health [80]. Participants also noted that musical preferences vary widely in South Africa, with marked differences based on age, region, ethnic group, and individual. As one participant explained:

*"The type of a song …that must unify everyone…It will never happen. It will be impossible."* – Music expert, man

While emphasising the diversity of musical preferences and experiences, our discussions also highlighted features of music that are conducive to facilitating active participation in a group setting. These included features such as cyclical, repetitive forms, and dense textures and timbres that are commonly found in participatory music styles as described by ethnomusicologist Thomas Turino [76]. As one participant explained,

*"If the repertoire is not known, then the tunes should be easy enough to learn for somebody that doesn't know. And also this is another aspect where indigenous music in this region already accommodates for that by lining up… response, cyclic repetitive forms. They are already built for community participation."* – Music expert, woman

In contrast, participants suggested that using European choral music, with its associated scores and required note-reading, would exclude participants without music literacy or stylistic familiarity, and create an environment less conducive to comfortable participation. One participant explained,

*"People don't want things that are too technical like ok let's look at these notes, then that changes the vibe."* – NGO Management, woman

*"Like a White European church… setting is not…is not ideal except for those people that grew up in that setting"* – Music expert, man

Participants highlighted the need for a flexible and adaptable approach to music making to encourage people to feel comfortable and safe to participate and connect with others. While potential benefits of music participation are emphasised in existing research [81–83], our participant responses highlight the importance of understanding cultural norms of music participation in order to create inclusive and accessible musical spaces; this is a topic that deserves greater attention from researchers [84].

The interplay between musical features and lyrical messages was also discussed. Participants noted that song lyrics can be supportive of mental health by sharing messages of affirmation and encouragement, as well as information about coping strategies. At the same time, discussions highlighted a potential tension between collective participation in music making and overly complex lyrical messages. As one participant explained:

*"The more fundamental way of using music which uses the material of what musicing is that itself; it has the potential to bring people together to have an experience, because it's a shared experience… you don't have to have very direct messages… for people to overcome particular problems…"* – Music Educator, woman

Participants in the music expert focus group recommended that a foundation of accessible, inclusive music making should be the first priority, with lyrical messages incorporated in such a way that they support comfortable participation of the group in the music. A further consideration is that the song lyrics should be easily understood for listeners, and sensitive to local understandings of perinatal mental health and appropriate forms of support.

Discussions also highlighted the need to consider the social relationships within music making and to identify appropriate facilitators. Participants from the health worker and traditional healer focus groups felt that the ideal facilitators of a music making intervention to support mental health would ideally possess both musical and clinical skills. Participants

from the other focus groups placed more emphasis on the musical aspects of the facilitator role and the importance of their social position in the community. Most importantly, facilitators should be able to engage the ingredients of music described above to make people feel comfortable and safe participating. This requires a shared musical language, but most importantly an emphasis on building social connectedness and enjoyment through group music making, rather than a specific idea of musical perfection [85].

**Contexts of music making.** The second sub-theme in this category describes the way making music to support mental health requires consideration of the contexts of music making as well as the ingredients of music itself. The social contexts and spaces where music making occurs shapes the way in which it is experienced by participants and the extent to which it fulfils the potential to build social connectedness, lift mood, and encourage and motivate women during the perinatal period.

Discussions highlighted the need to consider carefully what meeting places would be conducive to participation in music making in the local setting. Potential settings that were discussed included clinics, hospitals, schools, churches, women's association meetings, savings schemes (stokvel), and other community meeting places.

Participants in the CHW focus group identified both challenges and benefits of connecting music programs with facility-based health services:

*"My initial feeling was the clinic is not ideal, in terms of how busy it is … even just getting to the clinic sometimes, depending on the location of the clinic, can be quite challenging. [But] …I know in our context …we found that as long as you partner with the clinic, good things can come from that. Trying to set something up completely separate, it often just doesn't get off the ground …as a route in, it could be great. And to try and keep that connection, in terms of space and busyness it might have to happen in a different location. But, not to lose that link, I think, is quite important." – Health Worker, woman*

For this health worker, the clinic was seen as a necessary starting point for organising and recruiting participants, but not always the ideal location for music making. Another participant noted that in some cases, running music programs integrated within health services may present challenges because of the widespread stigma associated with mental health problems. In some cases, non-medical community settings might be preferred to encourage participation and avoid problems of stigma.

Other participants noted that making music in clinical spaces can have benefits in making the space more friendly and comfortable for patients, as discussed in relation to theme 3.1 above:

*"Singing in a hospital setting is really powerful… to take over that space with your voices and make it your space again…the act of singing together creates a sense of community …in a space where they wouldn't have felt like they are in a community with each other." – Health Worker, woman*

The varied perspectives on the most suitable locations for making music to support perinatal mental health indicate that there is a need to consider the preferences and experiences in the particular context, recognising that some spaces may encourage participation from certain groups while functioning to exclude others. Overall, our discussions of musical features, facilitators, and spaces of music making highlighted the need for careful attention to culture and context in research on music and health [61,80].

There were some limitations of the research. Firstly, we did not include a group of perinatal women themselves. This was due to a lack of capacity within the scope of this small research project. To some degree, this was mitigated by the fact that the CHWs were representatives of this stakeholder group due to their being selected by the NGO from among mothers in the community that they serve. Secondly, although there was saturation of most of the data from each FGD, a more comprehensive dataset may have been obtained if more health professionals and NGO staff from other community-based organisations had been recruited.

Our FGDs highlighted the many challenges experienced by women during the perinatal period. Importantly, experiences of perinatal mental distress were seen to be strongly influenced by social and structural factors that may be outside the control of individual women and families. The discussions demonstrated a need for forms of support that are accessible and non-stigmatising, as well as the importance of carefully considering social contexts and challenges when designing an intervention for perinatal women. This research contributes to broader scholarly conversations about the importance of culture in the provision of mental health support [86], and the need to consider local perspectives and meanings of music making to ensure that programs are culturally sensitive and appropriate [80].

The research participants emphasised the way that music is integrated into many aspects of cultural life in South Africa, serving important, accessible social and spiritual roles. The diversity of musical practices in contemporary South Africa was highlighted, as were concerns about loss of cultural expertise over time, as younger people no longer learn music and healing practices from elders. This suggests that there is a need for new or revived approaches to sustain musical practices that support health. At the same time, our research shows that there is a strong precedent for employing music to support perinatal health, including in formal health care contexts as well as in family and community settings. This indicates that a music-based intervention for perinatal mental health may build upon prior established practices and is likely to be considered acceptable and culturally appropriate, especially when design considerations include the particularities of community context.

The FGDs also provided an overview of complex ideas about the benefits of music making in the South African context. These included understandings of the way music can support the development of strong social connections and serve as a tool to transform mood and spirit in the context of traditional healing practices. This was seen as particularly important for perinatal women who may be experiencing social isolation and stigma.

Tying these themes together, our discussions underscored important factors to consider when developing a music-based intervention to support perinatal mental health. These included attention to both the "ingredients of music" itself as well as the contexts in which music is made. It was considered essential to leverage the benefits of music to help address the challenges experienced by perinatal women, by fostering social connection in an enjoyable and supportive environment. The discussions emphasised the need for music to be accessible, participatory, and familiar to participants, with a focus on their comfort and enjoyment rather than achieving a particular musical outcome. Our research participants had different views on the ideal facilitators of a music intervention, suggesting that this may depend upon the particular setting and community needs. However, they agreed that the facilitators and contexts need to be familiar and comfortable to encourage participants, and that overly technical approaches would be unhelpful.

## Conclusions

This study explored how community music practices in South Africa could support perinatal mental health and guide development of a CHIME style intervention. Stakeholders described a vibrant musical culture where collaborative music-making and singing is part of social, spiritual and family life, including long standing traditions linked to pregnancy and childbirth. Many noted that some of these practices are becoming less commonly practiced, despite their fostering meaningful connection, calm and emotional expression.

Participants highlighted strong alignment between these musical traditions and the emotional and social needs of perinatal women who often experience poverty, isolation, stigma and stress. Music was viewed as accessible, enjoyable and able to create supportive spaces that reduce distress and strengthen social bonds. A community-based intervention was seen as acceptable if it draws on familiar participatory forms and simple uplifting songs. Stakeholders emphasised that co-design should centre community knowledge, responsive facilitation and culturally meaningful settings.

## Supporting information

**S1 Checklist. COREQ (COnsolidated criteria for REporting Qualitative research) checklist.**
(PDF)

**S1 Text. Focus group discussion guides.**
(DOCX)

**S2 Text. Positionality statement.**
(DOCX)

**S1 Table. Positionality table.**
(DOCX)

**S2 Table. Themes with example quotes.**
(DOCX)

## Acknowledgments

We would like to thank our participants for their valuable input in developing this work. An enormous thank you to Noma Moshani for her translation and transcription services. We are deeply grateful for One to One Africa for the opportunity and support to collaborate with them and their team of Mentor Mothers.

## Author contributions

**Conceptualization:** Siphumelele Sigwebela, Bonnie B. McConnell, Thandi Davies, Katie Rose M. Sanfilippo, Lauren Stewart, Vivette Glover, Simone Honikman.

**Formal analysis:** Siphumelele Sigwebela, Bonnie B. McConnell, Ncumisa Waluwalu.

**Funding acquisition:** Katie Rose M. Sanfilippo, Lauren Stewart.

**Investigation:** Siphumelele Sigwebela, Ncumisa Waluwalu, Thandi Davies, Simone Honikman.

**Methodology:** Siphumelele Sigwebela, Bonnie B. McConnell, Katie Rose M. Sanfilippo, Lauren Stewart, Simone Honikman.

**Project administration:** Sally Field.

**Supervision:** Simone Honikman.

**Writing – original draft:** Siphumelele Sigwebela, Bonnie B. McConnell.

**Writing – review & editing:** Bonnie B. McConnell, Thandi Davies, Katie Rose M. Sanfilippo, Lauren Stewart, Sally Field, Vivette Glover, Simone Honikman.

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
