## [Decision Letter · Decision Letter 0]

15 Oct 2025

PGPH-D-25-01381

Community Health Intervention through Musical Engagement (CHIME) in South Africa:  A formative investigation of the feasibility and development of a music-based intervention to support perinatal mental health

Dear Dr. Sanfilippo,

Thank you for submitting your manuscript to PLOS Global Public Health. After careful consideration, we feel that it has merit and would be published subject to minor amendments. Therefore, we invite you to submit a revised version of the manuscript that addresses the points raised during the review process.

EDITOR:

Both reviewers are complimentary and have suggested reasonable minor revisions.

We look forward to receiving your revised manuscript.

Kind regards,

Graeme Hoddinott, Ph.D

Academic Editor

Journal Requirements:

Additional Editor Comments (if provided):

Reviewers' comments:

Reviewer's Responses to Questions

**Comments to the Author**

1. Does this manuscript meet PLOS Global Public Health’s publication criteria?

Reviewer #1: Yes

Reviewer #2: Yes

2. Has the statistical analysis been performed appropriately and rigorously?

Reviewer #1: Yes

Reviewer #2: N/A

3. Have the authors made all data underlying the findings in their manuscript fully available (please refer to the Data Availability Statement at the start of the manuscript PDF file)?

Reviewer #1: Yes

Reviewer #2: Yes

4. Is the manuscript presented in an intelligible fashion and written in standard English?

Reviewer #1: Yes

Reviewer #2: Yes

Reviewer #1: The study is both timely and essential in the context of perinatal care, particularly when there is a growing need to refocus efforts toward improving the quality of perinatal mental health services, especially in sub-Saharan Africa. It introduces accessible and cost-effective interventions that are already embedded within community settings. If appropriately adopted and tailored to perinatal mental health, these interventions have the potential to significantly improve maternal health outcomes.

The title is generally informative. However, given the qualitative nature of the findings, the study appears to be more exploratory than investigative. I recommend rephrasing the title to reflect this, for example: “...a formative exploration of the…”

Additionally, the CHIME intervention should be clearly defined or operationalized, as it is currently not well explained.

• The description of the study site is minimal. I suggest the authors provide a more comprehensive overview of the study site, including relevant characteristics, to justify its selection.

The integration of results and discussion, has in this case limited the depth of discussion. The discussion is brief, lacks adequate citation of supporting or contrasting literature, and does not sufficiently highlight the implications of the findings. I recommend either separating the results and discussion sections or significantly enhancing the discussion under each theme to provide deeper analysis and context, ultimately supporting stronger recommendations.

6. Conclusion

The conclusion currently includes the study's limitations. These would be better placed in a separate section preceding the conclusion.

Additionally:

• The conclusion should focus more explicitly on the study’s aim not lengthy.

• Lines 588–589 include citations, which are not typically necessary in the conclusion. This section should instead synthesize key findings in relation to the study objectives.

see attachment

Reviewer #2: Thank you very much for the opportunity to review this really important manuscript. It was very well written, and outlines in excellent detail the development of a new approach to maternal mental health care for women in South Africa. I have very limited comments on how this manuscript could be strengthened.

1)

Line 98-116 : The application of CBPR is to be celebrated here. However, a primary aim of this, is actually transformation and translation of power and decision making within research process. This is not in alignment with what the study actually does - where the external aims and objectives of the project are pre-determined by external stakeholders. Can the authors reflect on this particular tension? How was this mediated by the research team? You make mention of using decolonial feminist frameworks to structure your approach - can you add detail on how this was actually achieved? Is it through analysis? the way questions are defined? Sampling? this detail is needed, and should appear in the methods section of the manuscript.

2. line 157: Your positionally section is very light touch. Intersectionality and decolonial feminist principles would imply a more robust detailing here about researchers - you talk about nationality, but not about any other salient aspects of identity that would contribute to us understanding your positions within the work (i.e. career stages; ethnicity or cultural traditions; personal experiences of motherhood etc). Some of this would be helpful. In line 169-173 you make mention of the co-faciliator role and how it enables senior academics to deviate from formal academia and reflect on their on biases - I think it would be useful to give a specific example of what this looked like in practice.

3. Line 184: Why were the codes not shared back with the research participants? in CBPR one would anticipate some form of member checking after the analysis, not just for ensuring the circularity of the research process but also to ensure that what was distilled through remained in line with what participants contributed to the process during focus groups. This is a more important form of internal validity, than what the authors refer to in the discussion about potential translation discrepancies - which are important, but not the only important form of validity here. It would also be useful to see a sample coding framework - with detail and examples of how codes and categories shaped each thematic area.

4. Line 207 - it is not sufficient to tell us that you gain new insights into intergenerational trauma etc. can you elaborate on what these are, or where we can anticipate to find these insights within the remainder of the work that follows? There is too much in this section defining what it is, rather than focusing on what reflexivity resulted in.

5. Line 572: In the discussion section, It is not customary to start with limitations. The study raises some interesting findings, and it would be more effective to have this come towards the end of the section.

6. The discussion section could be expanded slightly - are the findings about music and wellbeing in line with, or divergent from what existing evidence says? would the south African approach need to make things different in its delivery from other music interventions used in other contexts? what are the similarities and differences that seem to emerge? The point raised earlier about intergenerational trauma, for example - seems like it is something that should be revisited here - is this pointing us towards needed to more collective/family oriented delivery to acknoweldge that trauma is situated across entire family networks - that are often connected to women's caring resources (and thus their mental health outcomes/therapy opportunities?)

Addressing these few points would make this paper have a much wider relevance to the field, and I hope that my comments are useful in that regard for the authors.

**Do you want your identity to be public for this peer review?** For information about this choice, including consent withdrawal, please see our Privacy Policy

Reviewer #1: **Yes:** GLADYS NAKIDDE (BNS, MNS, PhD)

Reviewer #2: No

---

## [Decision Letter · Decision Letter 1]

22 Dec 2025

Community Health Intervention through Musical Engagement (CHIME) in South Africa:  A formative exploration of the feasibility and development of a music-based intervention to support perinatal mental health

PGPH-D-25-01381R1

Dear Dr  Sanfilippo,

We are pleased to inform you that your manuscript 'Community Health Intervention through Musical Engagement (CHIME) in South Africa:  A formative exploration of the feasibility and development of a music-based intervention to support perinatal mental health' has been provisionally accepted for publication in PLOS Global Public Health.

Best regards,

Graeme Hoddinott, Ph.D

Academic Editor

Reviewer Comments (if any, and for reference):

Reviewer's Responses to Questions

**Comments to the Author**

Reviewer #1: All comments have been addressed

publication criteria?

Reviewer #1: Yes

3. Has the statistical analysis been performed appropriately and rigorously?

Reviewer #1: N/A

4. Have the authors made all data underlying the findings in their manuscript fully available (please refer to the Data Availability Statement at the start of the manuscript PDF file)?

Reviewer #1: Yes

5. Is the manuscript presented in an intelligible fashion and written in standard English?

Reviewer #1: Yes

Reviewer #1: The author has addressesd the comments. Thank you for the opportunity to review

**Do you want your identity to be public for this peer review?** For information about this choice, including consent withdrawal, please see our Privacy Policy

Reviewer #1: **Yes:** GLADYS NAKIDDENone
